# Macrophage Depletion Alleviates Immunosenescence in Diabetic Kidney by Modulating GDF-15 and Klotho

**DOI:** 10.3390/ijms26093990

**Published:** 2025-04-23

**Authors:** Asma S. Alonazi, Rana M. Aloraini, Lama M. Albulayhi, Layal M. Alshehri, Anfal F. Bin Dayel, Maha A. Alamin, Nouf T. Aldamri, Tahani K. Alshammari, Dalal A. Alkhelb, Wedad S. Sarawi, Hanan K. Alghibiwi, Nawal M. Alrasheed, Doaa M. Elnagar, Nouf M. Alrasheed

**Affiliations:** 1Department of Pharmacology and Toxicology, College of Pharmacy, King Saud University, Riyadh 11451, Saudi Arabia; aaloneazi@ksu.edu.sa (A.S.A.); 439200040@student.ksu.edu.sa (R.M.A.); 438202420@student.ksu.edu.sa (L.M.A.); 441200560@student.ksu.edu.sa (L.M.A.); abindayel@ksu.edu.sa (A.F.B.D.); mahaali@ksu.edu.sa (M.A.A.); naldamri@ksu.edu.sa (N.T.A.); talshammary@ksu.edu.sa (T.K.A.); dalkhelb@ksu.edu.sa (D.A.A.); wsarawi@ksu.edu.sa (W.S.S.); halghibiwi@ksu.edu.sa (H.K.A.); nalrasheed@ksu.edu.sa (N.M.A.); 2Department of Zoology, College of Science, King Saud University, Riyadh 11451, Saudi Arabia; elnagard1@yahoo.com; 3Department of Zoology, Faculty of Woman, Ain Shams University, Cairo 11566, Egypt

**Keywords:** growth/differentiation factor-15, Klotho, diabetic nephropathy, liposomal chlodronate, streptozotocin, immunosenescence

## Abstract

Cellular senescence is a hallmark of aging and contributes to age-related diseases, including diabetic nephropathy (DN). Additionally, macrophage-mediated inflammation has been linked with DKD. Therefore, we investigated the effect of macrophage depletion on kidney cell senescence in DN, focusing on the relationship between the GDF-15 and Klotho signaling pathways. Wistar albino rats (*n* = 24) were divided into four groups: healthy control, liposomal clodronate (LC)-treated healthy, diabetic, and LC-treated diabetic groups. Rats in the LC-treated healthy, diabetic, and LC-treated diabetic groups were intravenously administered LC once a week for 4 weeks. Rat models of type 2 diabetes were successfully established via the administration of streptozotocin and a high-fat diet, as evidenced by increased blood glucose levels, kidney weight to body weight (KW/BW) ratio, serum albumin, creatinine, and urea levels, kidney damage, and oxidative stress. However, LC-mediated macrophage depletion reduced the KW/BW ratio, improved serum and oxidative parameters, decreased inflammatory markers (IL-6 and TNF-α), and ameliorated oxidative stress. Additionally, LC treatment promoted macrophage polarization towards the anti-inflammatory phenotype, downregulated GDF-15 expression, upregulated Klotho expression, and ameliorated kidney damage. In conclusion, macrophage depletion combats kidney senescence by modulating Klotho and GDF-15, indicating their potential as novel targets in DN treatment.

## 1. Introduction

Diabetic nephropathy (DN), or diabetic kidney disease (DKD), is one of the most severe complications. It is defined as chronic kidney disease and a major cause of end-stage renal disorder in diabetic patients [1]. DN is characterized by pathological levels of urine albumin, diabetic glomerular lesions, and a loss of glomerular filtration rate (GFR) in patients with diabetes. DN is commonly associated with type 2 diabetes mellitus (T2DM), insulin deficiency, and insulin resistance [2].

Cellular senescence is a process that globally regulates cell death and is a hallmark of aging [3]. Individuals with T2DM and older nondiabetic individuals have increased senescence in kidney cells [4,5]. Research findings indicate that macrophages surround senescent cells during murine development and that macrophage infiltration leads to senescent cell clearance and promotes tissue remodeling in kidney injury or damage. Therefore, macrophage infiltration may be involved in kidney senescence in DN [6,7].

Macrophages act differently depending on whether they have experienced early or late acute kidney injury (AKI). Notably, macrophage infiltration increases markedly in the first 48 h after ischemic AKI and in the later phases. M2-like macrophages exhibit beneficial effects, such as intraluminal debris clearance, promotion of epithelial regeneration, regulatory T cell activation, and attenuation of kidney inflammation after ischemic AKI [7,8]. Infiltrating macrophages in the early stages of chronic kidney disease (CKD) are mainly of the M1 phenotype, with the proportion of infiltrating M2 cells increasing as the disease progresses. However, there is an increase in the proportion of macrophages with a high degree of autophagy at a later stage in mouse models, which could be attributed to an increase in the proportion of M2 cells [4,8]. Additionally, emerging evidence indicates that macrophage depletion following intraperitoneal injection with clodronate-containing liposomes improves proteinuria and renal function in db/db mice [9,10].

Importantly, the ability of macrophages to robustly clear senescent cells from body tissues diminishes with age, possibly because they acquire senescent-like properties, termed senescent-associated macrophages (SAMs) [6]. Senescent cell accumulation is common in renal injury, possibly due to Klotho downregulation, cyclin-dependent kinase inhibitor upregulation, telomere shortening, and oxidative stress [5]. Targeting senescence may be an effective strategy for ameliorating kidney injury in animal models and shows promise in humans. However, it is critical to distinguish between deleterious and protective senescence, indicating the need for further animal model research and clinical trials. Overall, it is anticipated that technological and research advances will enable the manipulation of this fundamental process [5].

Macrophages from the kidneys of mice with diabetes are deficient in growth/differentiation factor-15 (GDF-15), which is required for oxidative metabolism in M2-like macrophages, resulting in a decrease in Klotho, an antiaging single-pass membrane protein that protects against AKI [11,12].

Diabetes induces accelerated premature kidney aging and senescence [13], which is related to replicative stress-induced telomere shortening, resulting in cellular senescence [6]. This process is defined as irreversible cell cycle arrest (CCA), which is characterized by alterations in chromatin organization and gene expression that induce profound phenotypic changes. Senescent cells possess a specific secretome known as the senescence-associated secretory phenotype (SASP), which is enriched in proinflammatory cytokines, growth factors, and profibrotic proteins [7].

GDF-15, also known as macrophage inhibitory cytokine-1, is a divergent member of the transforming growth factor β superfamily. Under normal physiological conditions, its expression is low in most somatic tissues. However, GDF-15 is upregulated under stress or pathological conditions and is notably abundant in the placenta [14]. Due to its dynamic expression in disease states, GDF-15 has emerged as a potential biomarker of DN. Increased GDF-15 blood levels are linked to the development of T2DM, exacerbation of microalbuminuria, and progression of albuminuria in patients with T2DM. Considering that kidney injury is characterized by microalbuminuria, GDF-15 may significantly impact DN [15,16].

GDF-15 preserved Klotho expression during acute kidney injury and fibrosis, indicating a relationship between GDF-15 and Klotho expression [11]. Overall, these findings indicate that GDF15 drives a network of nephroprotective factors, including Klotho [17]. Klotho, a membrane-bound soluble protein, is associated with aging [18]. An in vitro experiment on the relationship between soluble Klotho levels and DN demonstrated that soluble Klotho levels were markedly lower in the early stages of DN, suggesting that soluble Klotho may serve as a novel DN biomarker [19,20,21].

Despite limited findings implicating diabetes in accelerated kidney senescence, regardless of biological aging-associated diseases, accumulating evidence strongly suggests a link between macrophage-mediated inflammation and DN [22]. Additionally, targeting macrophages and related signaling pathways has recently gained considerable research attention [23,24]. Based on these findings, we investigated the impact of macrophage depletion on kidney cell senescence in DN, focusing on the relationship between GDF-15 and soluble Klotho signaling pathways.

## 2. Results

### 2.1. Effect of Macrophage Depletion on Metabolic and Kidney Biomarkers in Rat Models of Streptozotocin/High-Fat Diet (STZ/HFD)-Induced Nephropathy

Serum glucose levels were significantly higher (*p* < 0.001) in the diabetic group (443.5 ± 34.14 mg/dL) than in the healthy control group (74.84 ± 4.20 mg/dL; Table 1). However, liposomal clodronate (LC) treatment significantly decreased (*p* < 0.001) serum glucose levels (89.53 ± 14.03 mg/dL) compared with the diabetic group (443.5 ± 34.14 mg/dL). Similarly, the final body weight was significantly higher (*p* < 0.001) in the diabetic group (327.2 ± 9.68 g) than in the control group (270.3 ± 7.35 g). However, there was no significant difference in final body weight between the LC-treated group (306.5 ± 11.16 g) and the diabetic model group (327.2 ± 9.68 g). Although rats in the diabetic model group (7.87 ± 0.42 mg/g) had a significantly higher (*p* < 0.001) kidney weight to body weight ratio (KW/BW ratio) than those in the control group (2.70 ± 0.37 mg/g), LC treatment significantly decreased the ratio to 2.93 ± 0.29 mg/g compared with the diabetic group (Table 1 and Appendix A).

Compared with the control group, there was a significant increase (*p* < 0.01) in serum albumin (4.25 ± 0.24 vs. 6.55 ± 0.40 g/dL), creatinine (2.73 ± 0.43 vs. 6.74 ± 1.10 mg/dL), and urea (5.30 ± 0.51 vs. 19.00 ± 2.37 mg/dL) levels and blood urea nitrogen (BUN) levels (2.62 ± 0.21 vs. 9.86 ± 0.68 mg/dL) in the diabetic group (Table 1). However, LC treatment significantly decreased serum albumin, creatinine, and urea levels to 3.62 ± 0.64, 2.04 ± 0.19, and 6.96 ± 1.34 mg/dL, respectively, compared with the diabetic model group (Appendix A). Additionally, LC treatment significantly decreased the BUN level to 2.87 ± 0.59 mg/dL compared with the diabetic model group.

### 2.2. Effect of Macrophage Depletion on the Levels of Oxidative Stress Biomarkers in Kidney Tissue of Rats with STZ-Induced Diabetes

Compared with the control group (47.3 ± 3.1 U/g), there was a significant increase (*p* < 0.001) in malondialdehyde (MDA) levels in the diabetic group (149.2 ± 5.70 U/g; Figure 1A). However, LC treatment significantly decreased (*p* < 0.001) MDA levels to 48.29 ± 1.46 U/g compared with the diabetic model group (Appendix A). Compared with the control group, there was a significant decrease (*p* < 0.001) in superoxide dismutase (SOD; 91.3 ± 1.8 vs. 25.53 ± 3.15 U/mg) and glutathione peroxidase (GPx; 25.27 ± 3.84 vs. 7.05 ± 0.75 U/mg) levels in the diabetic group. However, LC treatment significantly increased (*p* < 0.01) SOD and GPx levels to 62.39 ± 2.88 and 22.4 ± 1.40 U/mg, respectively, compared with the diabetic group (Figure 1B,C; Appendix A and Table 1).

### 2.3. Effect of Macrophage Depletion on Inflammatory Biomarker Levels in Kidney Tissue of Rats with STZ-Induced Diabetes

Compared with the control group, there was a significant decrease (*p* < 0.001) in IL-10 level (19.64 ± 0.41 vs. 7.68 ± 0.29 pg/mL) and a significant increase (*p* < 0.001) in IL-6 (4.96 ± 0.31 vs. 8.71 ± 0.27 pg/mL) and TNF-α (14.21 ± 1.39 vs. 26.57 ± 1.74 pg/mL) levels in the diabetic group (Figure 2A–C). However, LC treatment significantly increased IL-10 levels to 19.1 ± 0.60 pg/mL and decreased (*p* < 0.001) IL-6 and TNF-α to 4.19 ± 0.39 and 17.08 ± 0.32 pg/mL, respectively, compared with the diabetic model group (Figure 2A–C; Appendix A).

### 2.4. Effect of Macrophage Depletion on the Histopathology of Kidney Tissue and Macrophage Recruitment in Rats with STZ/HFD-Induced T2DM

Histological assay showed that the renal cortex of rats in the control group had a normal structure, glomerulus, proximal convoluted tubules, and normal distal convoluted tubules (Figure 3A). In contrast, the kidneys of rats in the model group showed a high incidence of macrophage infiltration, severe degeneration, and scattered inflammatory cells. Additionally, the kidneys of healthy rats treated with clodronate displayed less structural degradation, a near-normal kidney structure, and fewer macrophages. Moreover, hematoxylin and eosin (H&E) staining revealed that the kidneys of the LC-treated diabetic rats showed numerous fusiform-like macrophages between the tubules and reduced degradation areas (Figure 3A,B; Appendix A). Furthermore, there was a significant increase (*p* < 0.001) in macrophage-positive cells in the diabetic model group. In contrast, LC treatment significantly decreased (*p* < 0.001) the proportion of macrophage-positive cells. Moreover, periodic acid–Schiff (PAS) staining showed decreased macrophage infiltration in the kidneys of healthy rats treated with clodronate (Figure 3C,D; Appendix A).

### 2.5. Effects of Macrophage Depletion on MCP-1 and P16^INK4a^ Expression in Rats with STZ/HFD-Induced T2DM

The MCP-1 expression level is commonly used as an indicator of macrophage infiltration. Compared with the control group, there was a significant increase (*p* < 0.001) in MCP-1-positive cells (45.00 ± 5.520 vs. 179.3 ± 22.95%) in the diabetic group (Figure 4A,B). However, LC treatment significantly decreased (*p* < 0.001) the rate of MCP-1-positive cells to 53.00 ± 15.15% compared with the diabetic group (Appendix A).

Additionally, we examined the expression of the senescence marker P16^INK4a^ (Figure 4C,D). P16^INK4a^ expression was significantly higher (*p* < 0.001) in the diabetic group (173.1 ± 9.54%) than in the control group (49.34 ± 8.083%). In contrast, P16^INK4a^ expression decreased significantly (*p* < 0.001) in the LC group (41.78 ± 2.45%) compared with the diabetic group (173.1 ± 9.541%). Notably, there was no significant difference in P16^INK4a^ expression between the healthy control and LC-treated healthy groups (Appendix A).

### 2.6. Effects of Macrophage Depletion on CD86 and CD163 Expression in Rats with STZ/HFD-Induced T2DM

Compared with the control group, there was a significant increase (*p* < 0.001) in the expression of the M1 macrophage marker CD86 (42.00 ± 5.568 vs. 166.2 ± 30.70%) and a significant decrease (*p* < 0.01) in the M2 macrophage marker CD163 (234.5 ± 28.51 vs. 72.83 ± 6.31%) in the diabetic group (Figure 5A–D). However, LC treatment significantly decreased CD86 expression (34.50 ± 3.25 vs. 166.2 ± 30.70%) and increased CD163 expression (259.3 ± 32.33 vs. 72.83 ± 6.31%) compared with the diabetic group (Figure 5A–D; Appendix A). Notably, there were no significant differences in the expression levels of the M1 and M2 markers between the control and LC-treated healthy control groups.

### 2.7. Effect of Macrophage Depletion on GDF15 and Klotho Expression in Rats with STZ/HFD-Induced T2DM

Compared with the healthy control group, there was a significant increase in GDF-15 expression (1.01 ± 0.00 vs. 6.27 ± 0.58-fold induction) and a significant decrease in Klotho expression (1.00 ± 1.83 vs. 0.40 ± 0.04-fold induction) in the diabetic group (Figure 6A–D). However, LC treatment significantly decreased GDF-15 expression (1.38 ± 0.28-fold induction) and increased Klotho expression (1.20 ± 0.11-fold induction) compared with the diabetic group (Figure 6A–D; Appendix A). Importantly, there was no significant difference in Klotho expression between the healthy control and LC-treated healthy control groups (Appendix A).

## 3. Discussion

Our study provides greater insight into the intricate interplay between macrophages, immunosenescence, and DN. Elevated glucose levels and advanced glycation end products (AGEs) enhance the expression of adhesion molecules, cytokines, and chemokines in podocytes, mesangial cells, and epithelial cells in the kidneys, leading to the recruitment and activation of macrophages [22]. Macrophages play a crucial role in perpetuating inflammation by responding to the senescence-associated secretory phenotype (SASP), resulting in chronic inflammation and tissue damage [6]. High glucose and AGE levels promote ICAM-1 expression in podocytes, mesangial cells, and epithelial cells, thereby facilitating the adhesion of circulating monocytes. ICAM-1 is a crucial molecule that facilitates the recruitment of renal macrophages in DN. Additionally, activated resident macrophages and renal parenchymal cells in DN release chemokines such as C-C motif chemokine ligands 2 and 5 (CCL2 and CCL5) and macrophage colony-stimulating factor 1 (CSF-1). Collectively, these chemokines induce the differentiation of circulating monocytes (Ly6Chigh and Ly6Clow monocytes) into infiltrating macrophages within the kidneys, thereby contributing to the pathogenesis of kidney disease [22]. In response to the initial injury, resident macrophages detect damage-associated molecular patterns (DAMPs) and pathogen-associated molecular patterns (PAMPs), leading to augmented phagocytosis, antigen processing and presentation, and the secretion of proinflammatory cytokines. Moreover, bone marrow-derived monocytes are attracted to damaged tissues, where they mature into monocyte-derived macrophages, further intensifying the inflammatory response [25]. In addition to the physiological and pathological effects of senescent tissue cells on macrophages, senescent macrophages significantly influence tissue cells, establishing a bidirectional regulatory relationship between senescence and macrophages [26]. Notably, the interplay between senescent cells and macrophages amplifies the inflammatory process and contributes to tissue dysfunction and disease pathogenesis [27].

Importantly, this study revealed promising therapeutic avenues for mitigating DN progression by targeting macrophages in the kidney cells of patients with T2DM using LC. Notably, LC treatment caused significant improvements in DN biomarkers, including blood urea, BUN, serum albumin, and serum creatinine levels, indicating the amelioration of renal dysfunction. These biomarkers are essential indicators of kidney function and damage and highlight the potential of macrophage depletion as a therapeutic strategy for DN. Additionally, the observed alterations in oxidative stress biomarkers and inflammatory indicators further underscore the role of macrophages in mediating kidney damage in diabetes. In db/db mice, DN progression was linked to an increase in the number of kidney macrophages. Specifically, the accumulation and activation of macrophages in these mice correlated with elevated hyperglycemia, HbA1c levels, albuminuria, higher plasma creatinine levels, glomerular and tubular damage, renal fibrosis, and the expression of macrophage chemokines in the kidneys [28].

In this study, STZ combined with HFD was used to replicate the pathological features of T2DM and its associated complication, DN, in rats. This model was selected because of its ability to mimic the metabolic and inflammatory changes that are characteristic of human DN, including obesity, insulin resistance, and kidney dysfunction. Previous studies have demonstrated the suitability of the STZ/HFD model for investigating DN [29,30]. Barrière et al. reported that this combination effectively replicated the chronic complications of T2DM, including renal damage and inflammation, thus providing a robust platform for preclinical evaluation [31]. Genetic models are used less frequently because they are expensive and not fully representative of human T2DM. In contrast, the STZ/HFD model is cost-effective, easy to develop, and highly suitable for investigating the pathophysiological mechanisms of T2DM and for assessing potential therapeutic agents for the management of T2DM and its related complications [29,30,32]. STZ, first used in 1963, is the most common chemical used to induce diabetes and can be used to induce both type 1 and type 2 diabetes. High doses of STZ severely impair insulin secretion, similar to the characteristics of type 1 diabetes. In contrast, low doses of STZ cause a mild impairment of insulin secretion, which more closely resembles the later stages of T2DM. However, the low-dose STZ model does not address the insulin resistance commonly observed in patients with T2DM. Previous studies have shown that animals fed HFD develop insulin resistance, which further supports the use of a combined model. Incorporating HFD to induce peripheral insulin resistance, followed by a low dose of STZ to target pancreatic β-cells, would provide a more accurate representation of both the phenotype and the pathogenesis of human T2DM [33]. Additionally, the use of diets with varying fatty acid compositions, which are considered HFD, can introduce significant variability into the observed outcomes. Moreover, the use of extreme HFD may create a greater magnitude of tissue dysfunction or accelerate the process, making it impossible to study key aspects of the disease progression. Saturated and unsaturated fats differentially affect metabolic and immune responses in adipose tissue, promoting macrophage secretion of proinflammatory mediators, such as TNF-α, IL-6, and MCP-1 [34]. A study utilizing an STZ-induced diabetic CD11b-DTR mouse model demonstrated that macrophage depletion ameliorated albuminuria and preserved nephrin and podocin expression in diabetic kidneys, highlighting the direct role of macrophages in DN pathogenesis [35]. The ability of this model to simulate the interplay between macrophages and senescent cells is particularly valuable, as it allows us to explore the effects of macrophage depletion on immunosenescence in DN.

Furthermore, the observed shift in macrophage phenotypes in this study underscores the therapeutic relevance of macrophage modulation. M1 and M2 macrophages play opposing roles in renal inflammation. Significant improvements were observed in DN biomarkers and kidney histopathology after the depletion of proinflammatory M1 macrophages and the promotion of the anti-inflammatory M2 phenotype. This polarization probably reduces chronic inflammation and fosters a reparative environment in the kidneys. Injured renal epithelial cells release DAMPs and PAMPs, which activate endothelial cells to secrete chemokines, such as MCP-1 and ICAM-1. These chemokines facilitate the recruitment and migration of monocytes from the bloodstream to the kidneys. Tissue-resident macrophages recognize these molecular patterns through pattern recognition receptors (PRRs), which trigger their activation and cytokines [36]. These cytokines subsequently recruit additional macrophages to the bone marrow. This process promotes M1 macrophage polarization during the early stages of CKD [37]. In our study, macrophage depletion was achieved by administering LC for 4 weeks starting at 3 weeks post-diabetes induction. Notably, this regimen was implemented during the early stages of DN, a period when M1 macrophage polarization had already been initiated. M1 macrophages display high cell surface expression of CD16, CD32, CD80, and CD86, major histocompatibility complex class (MHC) II and IL-1 receptor (IL-1R), secrete proinflammatory cytokines (TNF-α, IL-1β, and IL-6), and have a high expression of oxidative and tissue-remodeling proteins (inducible nitric oxide synthase [iNOS], matrix metalloproteinases [MMPs], and macrophage-inducible C-type lectin), contributing to the activation and differentiation of fibroblasts into myofibroblasts and promoting the progression of fibrosis [38]. The release of inflammatory cytokines establishes a feedback loop that recruits more immune cells, reinforcing the inflammatory environment in the kidney. Additionally, circulating monocytes (CD11b+Ly6Chigh) are recruited to the kidneys, where they differentiate into proinflammatory M1 macrophages. As CKD progresses, there is a shift toward the M2 macrophage phenotype. M2 macrophages are induced by IL-4 and IL-13, contributing to the suppression of inflammation and the promotion of tissue repair and fibrosis. M2 macrophages express markers such as CD206, CD163, arginase-1 (Arg-1), and mannose receptor (MR) and secrete anti-inflammatory cytokines, primarily IL-10 and TGF-β, which help in tissue regeneration and fibrosis mitigation. Although M2 macrophages play a role in tissue repair, they also contribute to fibrosis by promoting collagen deposition through the release of VEGF and EGF [39]. Circulating Ly6Clow monocytes are recruited to the kidneys, where they differentiate into M2 macrophages with anti-inflammatory and profibrotic roles. Similarly, previous studies have demonstrated that M2 macrophages play a role in tissue repair and fibrosis mitigation, suggesting that targeting macrophage polarization could provide valuable insights into DN therapy [40,41,42]. Furthermore, hyperglycemia contributes to glomerular damage by decreasing the expression of renal proximal tubule AT1 receptor-associated proteins (ATARP), resulting in a decrease in the accumulation of tubulointerstitial M2 macrophages in diabetic kidneys [22,40]. The role of Toll-like receptors (TLRs) in inflammation has been studied in animal models and in patients with chronic CKD. The activation of TLR2 in macrophages triggers a proinflammatory response that contributes to nephropathy in diabetic mice. However, inhibition of TLR2 signaling in macrophages suppresses DN progression. Additionally, the downregulation of TLR4 and its signaling pathway promotes a shift in macrophage polarization from the proinflammatory M1 phenotype to the anti-inflammatory M2 phenotype, which helps reduce renal interstitial fibrosis, glomerulosclerosis, and loss of kidney function [43].

Moreover, this study demonstrated that LC-induced macrophage depletion effectively modulated the expression of GDF-15 and Klotho, which are pivotal for regulating immunosenescence and DN. A recent study reported that macrophages produce high levels of GDF-15 in response to oxidative and lysosomal stress, which can contribute to tissue fibrogenesis and angiogenesis [44]. Although Klotho is primarily expressed in the kidneys, it is also present in other tissues and cell types, including monocytes and macrophages. Therefore, modulating Klotho production could represent a potential pathway for regulating inflammatory response [45].

GDF-15, a stress-response cytokine and aging biomarker, is typically upregulated during DN and is associated with oxidative stress and inflammation. A previous study showed that the GDF-15 level was significantly higher in patients with DN than in those with diabetes without nephropathy and was associated with impaired renal function [46]. GDF-15 and other proinflammatory factors, such as TNF-α, IL-1β, and IL-6, were upregulated in the serum of rats with HFD/STZ-induced T2DM. Notably, elevated GDF-15 levels serve as a predictive indicator of DN [45]. In a high-glucose environment, podocytes activate the IKK/NF-κB signaling pathway. Ke et al. observed elevated receptor activator of NF-κB (RANK) expression in the podocytes of patients with DN, contributing to podocyte injury [47]. Importantly, the absence of GDF-15 worsens kidney damage, as GDF-15 helps reduce macrophage activation by inhibiting the NF-κB pathway [48].

In contrast, Klotho is an antiaging protein that exerts protective effects on the kidneys by mitigating inflammation and oxidative damage. Research indicates that Klotho-deficient mice with STZ-induced diabetes exhibit poor kidney function [49]. Donate-Correa et al. observed reduced Klotho levels in adults with T2DM, and low Klotho levels were correlated with high levels of inflammatory markers [50]. In vitro and in vivo experiments have shown that Klotho treatment effectively reduced high glucose-induced inflammation, oxidative stress, and cell death [51]. Li et al. demonstrated that Klotho overexpression mitigated high glucose-induced glomerular endothelial cell injury in db/db mice by inhibiting the Wnt/β-catenin signaling pathway and the renin–angiotensin–aldosterone system (RAAS) [52]. Collectively, these findings indicate that renal injury in various kidney disorders, including DN, induces the activation of the Wnt/β-catenin signaling pathway and the RAAS [53]. Additionally, Klotho overexpression protected against podocyte apoptosis by inhibiting oxidative stress via activation of the Nrf2 signaling pathway in DN. Nrf2 serves as a critical transcription factor in the regulation of cellular redox homeostasis [54]. Recent studies demonstrated that Klotho could regulate macrophage polarization toward the anti-inflammatory M2 phenotype by inhibiting the TLR4/MyD88/NF-κB pathway, thereby decreasing proinflammatory cytokines (IL-1β, TNF-α, IL-6) and enhancing anti-inflammatory factors (IL-10, TGF-β, CD206) and facilitating tissue repair [55]. Collectively, these findings confirm the renoprotective effects of Klotho in DN.

Macrophage depletion decreased proinflammatory marker levels, and GDF-15 significantly increased Klotho expression. Macrophages exhibit an inverse relationship with Klotho levels, as macrophages drive inflammation that downregulates Klotho expression. Conversely, Klotho can mitigate the inflammatory response by reversing macrophage senescence, which in turn alleviates the inflammatory burden and increases Klotho levels [52]. Overall, these findings suggest that the modulation of GDF-15 and Klotho is closely associated with macrophage depletion and plays a role in reducing senescence-associated inflammation and oxidative stress. Similarly, Valiño-Rivas et al. confirmed the protective role of Klotho in kidney injury and its inverse relationship with GDF-15. Although GDF-15 and Klotho typically have an inverse relationship under stress conditions, GDF-15 acts as a compensatory factor to counteract Klotho deficiency during kidney stress, suggesting that GDF-15 could indirectly support Klotho under certain conditions [11].

At the clinical level, the role of oxidative stress and proinflammatory factors in the pathogenesis of DN is evident [56]. Additionally, elevated levels of GDF-15 predict the mortality rate in DN patients with cardiac complications [15]. In support of this, our findings open new avenues for exploring the role of GDF-15 and Klotho in the progression and management of high-risk DN patients. These results could be clinically translated into a multimarker informative approach in DN patients. Oxidative stress–proinflammatory combination markers could predict the disease severity.

To the best of our knowledge, this study is the first to elucidate the interplay between macrophage depletion, GDF-15 (aging biomarkers), and Klotho (an antiaging protein) in mitigating immunosenescence and improving renal function. Notably, it may be possible to delay kidney senescence, improve renal function, and mitigate chronic inflammation and oxidative stress that drive DN progression by depleting macrophage levels, reducing GDF-15 expression, and restoring Klotho levels. These findings reinforce the importance of targeting immunosenescence pathways as part of a comprehensive therapeutic approach for DN. Additionally, the results provide a foundation for developing innovative therapies targeting macrophages and their downstream signaling pathways and emphasize the broader implications of immunosenescence modulation in treating age-related diseases.

### Limitations and Future Directions

Despite the advantages of the STZ/HFD model, it is important to note its limitations. For example, the rapid induction of stress factors and the lack of genetic heterogeneity may not fully represent the complexity of human DN. Therefore, further research is needed to elucidate the underlying mechanisms and determine the long-term effects of macrophage depletion on DK progression in humans. To enhance the translatability of findings to clinical settings, future studies should aim to address these limitations by incorporating additional variables or exploring alternative models. Future studies should consider assessing urinary albumin levels and applying functional parameters, such as glomerular filtration rate and creatinine clearance, to enhance the clinical relevance of the findings. Although this study employed repeated clodronate liposome dosing within a chronic DN model, it was limited to assessing the short-term effects of macrophage depletion using clodronate liposomes in DN. Accordingly, future investigations are needed to evaluate the long-term consequences and safety profile of sustained macrophage depletion, including its potential effects on tissue repair, paradoxical inflammation, and susceptibility to infection. Additionally, although our in vivo model revealed a clear association between macrophage depletion and the modulation of GDF-15 and Klotho expression, mechanistic experimentation using gene silencing or pathway-specific inhibitors remains limited in this context. Future studies employing in vitro models or transgenic models may help to clarify the regulatory mechanisms and confirm the causal relationships, particularly in relation to pathways such as NF-kB and Wnt/β-catenin. Moreover, future studies are warranted to evaluate the systemic immunomodulatory effects of clodronate liposome treatment by assessing circulating monocyte and leukocyte counts in blood, which could further validate the systemic impact of macrophage depletion beyond tissue-specific effects.

## 4. Materials and Methods

### 4.1. Drugs, Chemicals, and Antibodies

Streptozotocin (STZ) was purchased from Sigma-Aldrich (St. Louis, MO, USA). Clodronate liposomes (liposomes) were supplied by Liposoma BV (Science Park 408, Amsterdam, The Netherlands). Inflammatory biomarkers, including tumor necrosis factor-α (TNF-α), interleukin 10 (IL-10), and interleukin 6 (IL-6), were measured using specific enzyme-linked immunosorbent assay (ELISA) kits for rats from Solarbio Ltd. (Liandong, Beijing, China). Anti-GDF-15, anti-Klotho, CD86 (M1), CD163 (M2), adhesion molecule markers, monocyte chemoattractant protein-1 (MCP-1), and P-16 antibodies were obtained from Solarbio Ltd. (Liandong, Beijing, China). All other chemicals and reagents were of analytical grade and were purchased from standard commercial companies.

### 4.2. Experimental Animals

Male Wistar albino rats (8–10 weeks old) weighing 250–350 g were obtained from the Animal Care Center, College of Pharmacy, King Saud University, Saudi Arabia. The rats were appropriately housed and maintained in special cages under controlled conditions (temperature, 22–23 °C; humidity, 60%; and 12 h/12 h light/dark cycle). During the experiment, the animals were provided with distilled water and rodent maintenance chow ad libitum. Rats in the nondiabetic groups were fed a normal standard chow diet, whereas those in the diabetic groups were fed a high-fat diet (containing a mixed meal of 63% calories, 1% [*w*/*w*] sucrose, 1% [*w*/*w*] cholesterol, high levels of protein, and 25% extra-virgin olive oil), with some modifications to induce obesity and insulin resistance, which are considered hallmarks of type 2 diabetes. The study protocol was performed in accordance with the Experimental Animals Ethics Committee Act of the King Saud University Institutional Research Ethics Committee (KSU-SE-23-39). The regimen used in this study was based on preliminary studies conducted by other laboratories.

### 4.3. Induction of Diabetes

Type 2 diabetes was induced in the rats after a 24 h fast using multiple intraperitoneal injections of STZ dissolved in 0.1 M citrate buffer (pH 4.5) immediately prior to injection (30 mg/kg; three doses once per week) [29,31]. Blood samples were collected via the tail vein 72 h after STZ injection to measure blood glucose levels using the Contour TS blood glucose monitoring system (Roche Diagnostic, Indianapolis, IN, USA). Diabetes was confirmed when the blood glucose levels reached >11.1 mmol/L (200 mg/dL) [57]. Thereafter, rats were selected, included in this study, and allocated to the second and fourth groups.

### 4.4. Depletion of Macrophages

After the induction of T2DM in the rats, we investigated the effect of depleting kidney macrophages on inflammation in the context of DCM. Macrophages were chemically depleted using a liposomal clodronate solution and phosphate-buffered saline (PBS) liposomes (control). Briefly, the rats were anesthetized using CO_2_ and intravenously (i.v.) injected with liposomal clodronate (LC; 15 mg/kg) via the tail vein once per week for 4 weeks (four doses in total) starting from week 3 of the treatment period [58,59]. Notably, the liposome solution used for macrophage depletion was stored at 4–8 °C and transferred to room temperature 30 min before use.

### 4.5. Experimental Design

Twenty-four (12 nondiabetic and 12 diabetic) rats were weighed and allocated into four groups, with six rats in each group (*n* = 6). The rats were treated daily for six consecutive weeks as follows (Figure 7):**group 1:** Rats in the nondiabetic control group received normal saline (0.9% NaCl) via oral gavage starting from week 3 for a 4-week period.**group 2:** Diabetic rats in the untreated group received normal saline (0.9% NaCl) via oral gavage starting from week 3 for a 4-week period.**group 3:** Nondiabetic rats were treated with LC (15 mg/kg, i.v.) once a week starting from week 3 for a 4-week period [58,59].**group 4:** Diabetic rats treated with LC once a week starting from week 3 for a 4-week period.

The body weights of the rats were recorded weekly until the end of the experiment. At the end of the experiment, the rats were fasted overnight (12 h), sacrificed by gradually increasing the concentration of CO_2_, and decapitated. Blood samples were collected and processed to separate serum for biochemical analyses. Kidneys were immediately removed, rinsed with ice-cold PBS, and weighed. Additionally, the ratio of kidney weight to body weight (KW/BW) was calculated as an index of diabetic nephropathy. Thereafter, the kidney samples were homogenized in cold PBS (10% *w*/*v*) to collect a clear homogenate for the analysis of kidney biomarkers. Serum creatinine, albumin, urea, and blood urea nitrogen (BUN) levels were measured using colorimetric kits from Abcam (UDi, Europe, Langenhagen, Hannover, Germany). For histological and immunohistochemical assay, a portion of the kidney samples was dissected and fixed in neutral buffered formalin (4%). Additionally, the remaining portion was stored at −80 °C for molecular analyses.

### 4.6. Biochemical and Molecular Analyses

#### 4.6.1. Determination of Serum Glucose Levels

Serum glucose levels were measured using a glucose analytical assay kit (RayBiotech, Peachtree Corners, GA, USA; Cat no. MA-GLU-1) according to the manufacturer’s instructions.

#### 4.6.2. Determination of Diabetic Nephropathy Biomarkers

Kidney biomarkers, including serum albumin, creatinine, and urea levels, were measured using rat colorimetric kits from UDi (Langenhagen, Hannover, Germany), according to the manufacturer’s instructions. To determine albumin, creatinine, urea, and BUN levels, 1 mL of the working reagent was added to 10 µL of each standard sample and incubated for 5 min at 37 °C. Thereafter, the absorbance of the samples was measured against a reagent blank at 630 and 340 nm using a spectrophotometer reader (Biochrom Ltd., Cambridge, UK). Finally, albumin, creatinine, urea, and BUN levels were calculated using specific formulas and expressed as mg/dL.

#### 4.6.3. Assessment of Oxidative Stress Biomarkers

Oxidative stress was assessed by determining glutathione peroxidase (GPx), superoxide dismutase (SOD), and malondialdehyde (MDA) levels. GPx levels were determined using the method described by Moron et al. [60]. Briefly, 1 mL of kidney homogenate was mixed with 1 mL of 25% trichloroacetic acid (TCA) and centrifuged at 3000 rpm for 10 min at 4 °C to collect the supernatant (0.5 mL), which was mixed with 4.5 mL of Ellman’s reagent. Thereafter, the absorbance was measured at 412 nm using a spectrophotometer. MDA was estimated using the thiobarbituric acid (TBA) assay, as described by Ohkawa et al. [61]. Specifically, a mixture of 1 mL of 0.6% TBA, 2.5 mL of 20% trichloroacetic acid, and 500 μL of kidney homogenate was heated in a boiling water bath for 30 min, cooled, and centrifuged at 4 °C. Finally, the absorbance of the developed chromogen was measured at 535 nm. Additionally, SOD activity was estimated using the nitro blue tetrazolium method described by Delides et al. [62], which measures the degree of inhibition of the reaction at 430 nm. SOD, GPx, and MDA activities were expressed as U/mg protein, U/mg protein, and nmol/mg protein, respectively.

#### 4.6.4. Assessment of Inflammatory Biomarkers

Briefly, the levels of inflammatory markers were determined using rat-specific immunoassay ELISA kits (Solarbio Ltd., Liandong, Beijing, China), according to the manufacturer’s instructions. Kidney homogenate or serum samples were placed in 96-well plates precoated with monoclonal antibodies targeting TNF-α, IL-6, and IL-10. Finally, the absorbance was measured spectrophotometrically.

#### 4.6.5. Western Blot Analysis

To detect GDF-15 and Klotho protein expression, proteins were extracted from frozen kidney tissue samples via homogenization in ice-cold lysis buffer and RIPA buffer supplemented with an equal mix of protease inhibitor and phosphatase inhibitor cocktails. Protein concentrations were determined using a Direct Detect quantification analyzer (EMD Millipore Corporation, Billerica, Germany). Western blotting was performed according to the method described by Towbin et al. [63]. Briefly, 50 µg of protein samples were separated using sodium dodecyl sulfate polyacrylamide gel electrophoresis (SDS-PAGE) and transferred to polyvinylidene difluoride (PVDF) membranes (0.2 µm, Immun-Blot, Bio-Rad). After blocking with a 5% mixture of nonfat dry milk and bovine serum albumin solution for 1 h at room temperature, the membranes were incubated overnight at 4 °C with primary antibodies diluted in Tris-buffered saline Tween (TBST) buffer at a 1:1000 ratio for GDF-15 (rabbit polyclonal anti-GDF15) and Klotho (mouse monoclonal anti-KL). β-actin, diluted to a 1:2000 ratio, was used as the housekeeping loading control antibody. Thereafter, the membranes were washed and incubated with horseradish peroxidase-conjugated anti-rabbit or anti-mouse (1:15,000) secondary antibodies diluted in TBST buffer at room temperature for 1 h. Protein blots were developed using an enhanced chemiluminescence detection kit (GE Healthcare, Buckinghamshire, UK) 2 min prior to image acquisition. Immunoreactive bands were visualized using the LI-COR Odyssey imaging system (Lincoln, NE, USA). The intensities of the different protein bands were quantified densitometrically using ImageJ software version 1.54 (NIHI Image, Bethesda, MD, USA) and normalized against the loading control (β-actin) by dividing the target protein value by the β-actin value. Relative values were normalized to the control value, which was arbitrarily fixed at 1 and assigned as a fold induction. Western blot analysis was performed on kidney lysates from six biological replicates per group (*n* = 6). Notably, one representative immunoblot per protein is presented in the main figures, while full unprocessed replicates are provided in Appendix A.

#### 4.6.6. Histological Examination

After inducing anesthesia using CO_2_, the kidneys of the rats were removed immediately, washed with ice-cold saline, and carefully cleaned of extraneous fat and connective tissue. Thereafter, the kidney samples were fixed in 10% neutral formalin for 24 h, dehydrated with high concentrations of ethanol, cleared with xylene, and embedded in paraffin. Paraffin-embedded sections (4 µm) were prepared and stained with hematoxylin and eosin (H&E) or periodic acid–Schiff (PAS) for morphological examination. Images were analyzed by investigators specializing in this field.

#### 4.6.7. Immunohistochemistry

An immunohistochemical assay was performed to detect the expression of macrophage markers (CD86, M1; CD163, M2) and adhesion molecule markers (MCP-1 and P16^INK4a^) in the kidney samples using specific kits, according to the manufacturer’s instructions (Advanced Technology & Industrial Co., Ltd., Shenzhen, China). Immunostaining was performed in accordance with the procedure described in a previous study [64,65] using the ImmunoCruz ABC staining system (Santa Cruz, Dallas, TX, USA). After blocking the nonspecific binding of the antibodies by incubating with the blocking reagent for 5 min, the sections were incubated with the primary antibodies overnight at 4 °C, washed three times in Tris buffer, and incubated with biotinylated anti-rabbit and anti-mouse IgG (1:100 dilution both) for 30 min. Thereafter, the samples were washed, followed by incubation with a diaminobenzidine substrate working solution and staining with Mayer’s hematoxylin. The sections were mounted in a mixture of distyrene, plasticizer, and xylene to preserve the staining. Finally, the tissue sections were observed using a bright-field light microscope (DMRBE; Leica, Bensheim, Germany) equipped with a video camera (ProgRes; Kontron Instruments, Watford, UK). Images were viewed and analyzed using ImageJ software.

### 4.7. Statistical Analysis

All data were expressed as mean ± standard error of the mean (SEM). Statistical analyses were conducted using Prism Windows Version 5 (GraphPad Software Inc., San Diego, CA, USA). Significant differences between groups were determined using one-way analysis of variance (ANOVA), followed by a Tukey–Kramer post hoc test. Statistical significance was set at *p* < 0.05.

## 5. Conclusions

This study provides valuable insights into the role of macrophage depletion in combating diabetes-accelerated kidney senescence through modulation of the GDF-15 and Klotho signaling pathways. Conclusively, the findings of our study have important implications for developing novel therapeutic strategies for DN and other age-related diseases associated with immunosenescence. Further research is warranted to translate these findings into clinical practice and to improve outcomes in patients with DN.

## Figures and Tables

**Figure 1 ijms-26-03990-f001:**
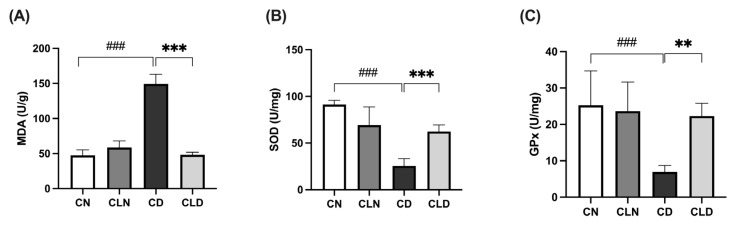
Effects of macrophage depletion on oxidative stress biomarker levels in kidney tissue of rats with streptozotocin/high-fat diet (STZ/HFD)-induced nephropathy. Data are expressed as mean ± standard error of the mean (SEM; *n* = 6 samples per group; each sample represents a different rat). Group comparisons were performed using one-way ANOVA, followed by a Tukey–Kramer post hoc test. ** *p* < 0.01 and *** *p* < 0.001 indicate statistical significance compared with the diabetic model group; ^###^
*p* < 0.001 indicate statistical significance compared with the healthy control group. (**A**) Malondialdehyde [MDA]; (**B**) superoxide dismutase [SOD]; (**C**) glutathione peroxidase [GPx]. Abbreviations: CN, normal control group; CD, diabetic control group; CLN, liposomal clodronate-treated nondiabetic group; CLD, liposomal clodronate-treated diabetic group.

**Figure 2 ijms-26-03990-f002:**
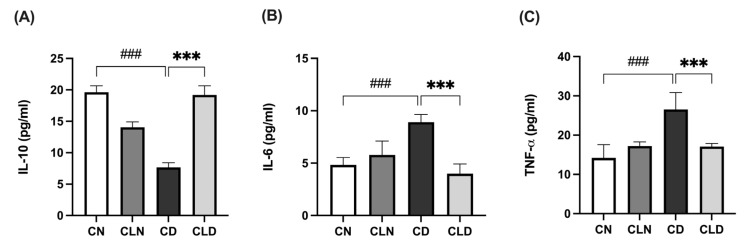
Effects of macrophage depletion on inflammatory biomarker levels in kidney tissue of rats with streptozotocin/high-fat diet (STZ/HFD)-induced nephropathy. Data are expressed as mean ± standard error of mean (SEM; *n* = 6 samples per group; each sample represents different rats per group). Group comparisons were performed using one-way ANOVA, followed by a Tukey–Kramer post hoc test. *** *p* < 0.001 indicates statistical significance compared with the diabetic model group; ^###^
*p* < 0.001 indicates statistical significance compared with the healthy control group. (**A**) Serum level of interleukin-10 [IL-10]; (**B**) serum level of interleukin-6 [IL-6]; (**C**) serum level of tumor necrosis factor-alpha [TNF-α]. Abbreviations: CN, normal control group; CD, diabetic control group; CLN, liposomal clodronate-treated nondiabetic group; CLD, liposomal clodronate-treated diabetic group.

**Figure 3 ijms-26-03990-f003:**
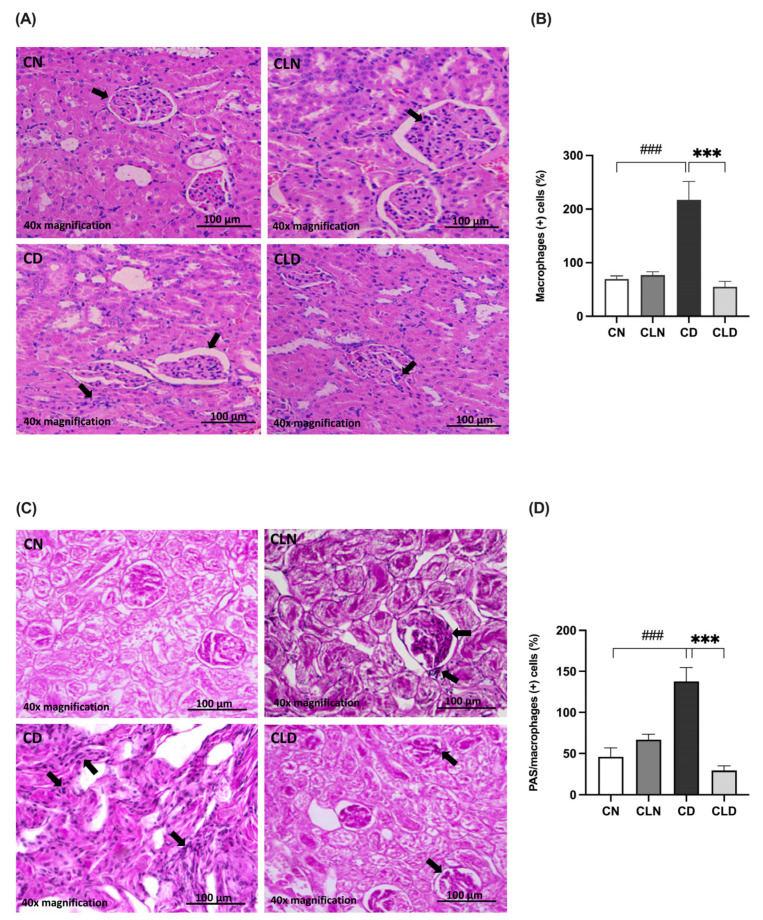
Assessment of histopathological changes in the kidneys of rats with streptozotocin/high-fat diet (STZ/HFD)-induced type 2 diabetes using hematoxylin and eosin (H&E) and periodic acid–Schiff (PAS) staining. (**A**,**B**) Photomicrographs of H&E-stained renal cortex of the rats, and quantification of the staining intensity. (**C**,**D**) PAS-stained renal cortex and quantification of macrophage-positive cells. Black arrows in (**A**) indicate cells with macrophage-like morphology observed in H&E-stained kidney section. In (**C**), black arrows point to PAS-positive cells with features consistent with infiltrating macrophages. Data are expressed as mean ± standard error of mean (SEM; *n* = 6 fields per slide; six slides per group). Group comparisons were performed using one-way ANOVA, followed by a Tukey–Kramer post hoc test. *** *p* < 0.001 indicates statistical significance compared with the diabetic model group; ^###^
*p* < 0.001 indicates statistical significance compared with the healthy control group. Magnification = 40×. Abbreviations: CN: normal healthy control group; CD: diabetic control group; CLN: liposomal clodronate-treated nondiabetic group; CLD: liposomal clodronate-treated diabetic group.

**Figure 4 ijms-26-03990-f004:**
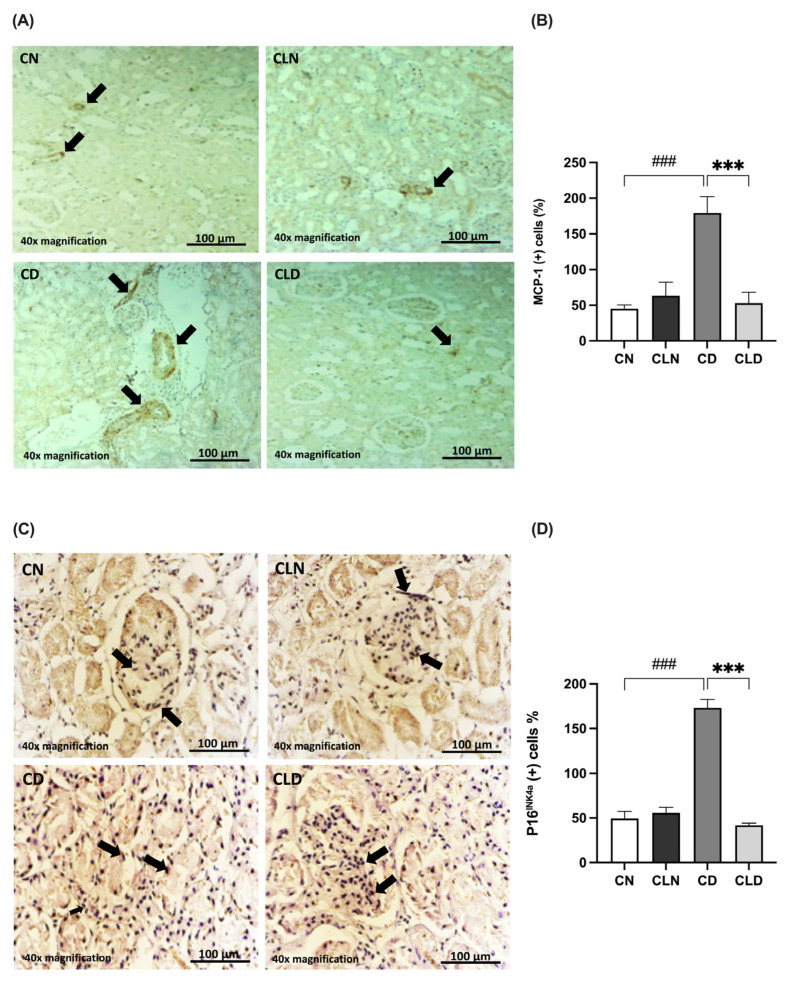
Effects of macrophage depletion on MCP-1 and P16^INK4a^ expression in rats with streptozotocin/high-fat diet (STZ/HFD)-induced type 2 diabetes. Photomicrographs of rat kidney sections stained with either MCP-1 antibody (**A**) or P16^INK4a^ antibody (**C**). Black arrows in (**A**) indicate MCP-1 positive cells identified by brown immunostaining in kidney sections. the black arrow in (**C**) highlight P16^INK4a^ Positive cells showing brown cytoplasmic or nuclear staining. (**B**,**D**) Quantification of MCP-1- and P16^INK4a^-positive cells. Data are expressed as mean ± standard error of the mean (SEM; *n* = 6 fields per slide, six slides per group; each slide represents kidney tissue section). Group comparisons were performed using one-way ANOVA, followed by a Tukey–Kramer post hoc test. *** *p* < 0.001 indicates statistical significance compared with the diabetic group; ^###^
*p* < 0.001 indicates statistical significance compared with the healthy control group. Magnification = 40×. Abbreviations: CN, normal control group; CD, diabetic control group; CLN, liposomal clodronate-treated non-diabetic group; and CLD, liposomal clodronate-treated diabetic group.

**Figure 5 ijms-26-03990-f005:**
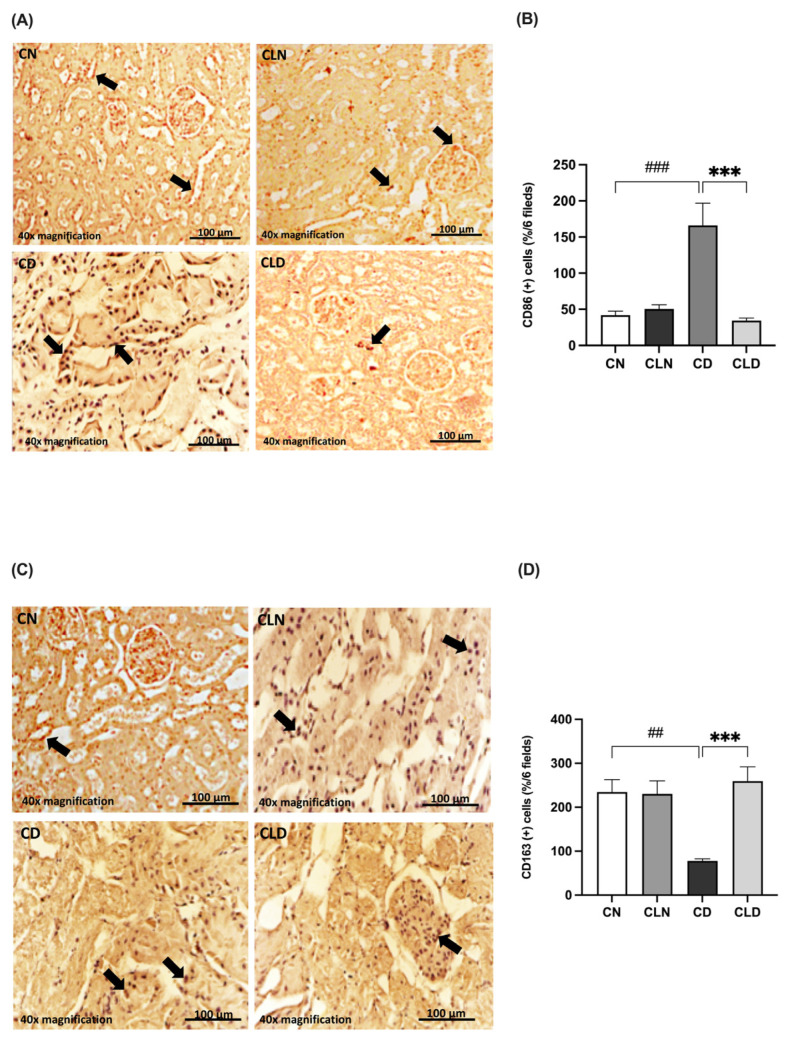
Effect of liposomal clodronate on the expression of M1 and M2 macrophage markers in rats with diabetic nephropathy. Photomicrographs of rat kidney sections stained with either CD86 antibody (**A**) or CD163 antibody (**C**). Black arrows in (**A**) indicate CD86-positive (M1) macrophages, while arrows in (**C**) highlight CD163-positive (M2) macrophages. (**B**,**D**) Quantification of CD86- and CD163-positive kidney cells. Data are expressed as mean ± standard error of the mean (SEM; *n* = 6 fields per slide, six slides per group; each slide represents kidney tissue section). Group comparisons were performed using one-way ANOVA, followed by a Tukey–Kramer post hoc test. *** *p* < 0.001 indicate statistical significance compared with the diabetic control group; ^###^
*p* < 0.001 and ^##^
*p* < 0.01 indicate statistical significance compared with the diabetic control group. Magnification = 40×. Abbreviations: CN, normal control group; CD, diabetic control group; CLN, liposomal clodronate-treated nondiabetic group; CLD, liposomal clodronate-treated diabetic group.

**Figure 6 ijms-26-03990-f006:**
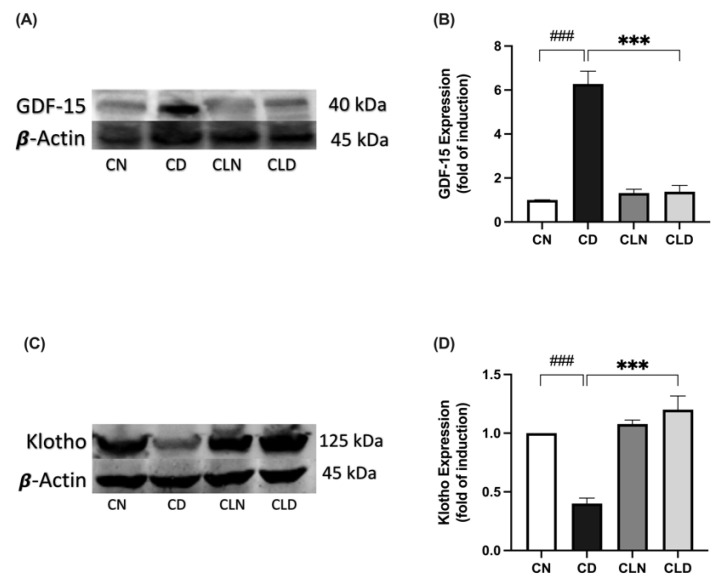
Effect of liposomal clodronate (LC) treatment on GDF-15 and Klotho expression in rats with streptozotocin/high-fat diet (STZ/HFD)-induced type 2 diabetes. Representative immunoblot of the GDF-15 antibody (**A**) or Klotho antibody (**C**) was selected from six biological replicates (*n* = 6/group). Full unprocessed immunoblot replicates are provided in Appendix A. (**B**,**D**) Quantification of GDF-15-1 and Klotho expression levels. Data are expressed as mean ± standard error of the mean (SEM; *n* = 6 samples per group; each sample from a different rat per group). Group comparisons were performed using one-way ANOVA, followed by a Tukey–Kramer post hoc test. *** *p* < 0.001 indicate statistical significance compared with the normal control group; ^###^
*p* < 0.001 indicates statistical significance compared with the diabetic model group, Magnification = 40×. CN, normal control group; CD, diabetic control group; CLN, liposomal clodronate-treated non-diabetic group; and CLD, liposomal clodronate-treated diabetic group.

**Figure 7 ijms-26-03990-f007:**
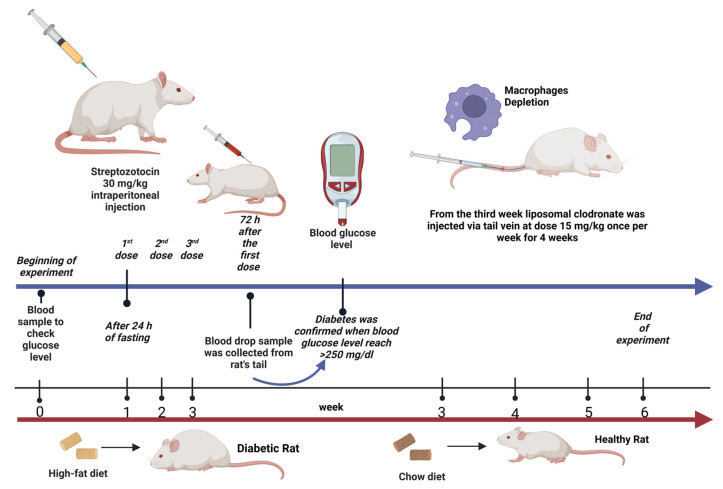
Experimental design illustration. Created with BioRender.com.

**Table 1 ijms-26-03990-t001:** Effect of macrophage depletion on metabolic and kidney biomarkers in a rat model of streptozotocin/high-fat diet (STZ/HFD)-induced nephropathy.

	Groups			
Parameters	CN	CLN	CD	CLD
**Glucose (mg/dL)**	74.84 ± 4.20	68.33 ± 4.82	443.5 ± 34.14 ***	89.53 ± 14.03 ^###^
**Final body weight (g)**	270.3 ± 7.35	249.7 ± 14.43	327.2 ± 9.68 **	306.5 ± 11.16
**KW/BW (mg/g)**	2.70 ± 0.37	3.29 ± 0.36	7.87 ± 0.42 ***	2.93 ± 0.29 ^###^
**Albumin (g/dL)**	4.25 ± 0.24	5.65 ± 0.29	6.55 ± 0.40 **	3.62 ± 0.64 ^###^
**Creatinine (mg/dL)**	2.73 ± 0.43	3.60 ± 0.37	6.74 ± 1.10 **	2.04 ± 0.19 ^###^
**Urea (mg/dL)**	5.30 ± 0.51	6.25 ± 1.31	19.00 ± 2.37 ***	6.96 ± 1.34 ^###^
**BUN (mg/dL)**	2.62 ± 0.21	2.67 ± 0.54	9.86 ± 0.68 ***	2.87 ± 0.59 ^###^

Data are expressed as mean ± standard error of the mean (SEM; *n* = 6 samples per group). Group comparisons were performed using one-way ANOVA, followed by the Tukey–Kramer post hoc test. ** *p* < 0.01 and *** *p* < 0.001 indicate statistical significance compared with the healthy control group; ^###^
*p* < 0.001 indicates statistical significance compared with that diabetic model group. Abbreviations: CN, normal control; CD, diabetic control; CLN, clodronate-treated normal; CLD, clodronate-treated diabetic; KW/BW, kidney weight/body weight; BUN, blood urea nitrogen.

## Data Availability

The study data are available upon request from the corresponding author.

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
