# Peer review of "Macrophage Depletion Alleviates Immunosenescence in Diabetic Kidney by Modulating GDF-15 and Klotho"

_ijms, 2025, doi:10.3390/ijms26093990_

Round 1
Reviewer 1 Report
Comments and Suggestions for Authors
This study investigated the impact of macrophage depletion on kidney cell senescence in diabetic kidney disease (DKD) in a rat model, with particular emphasis on the roles of GDF-15 and Klotho signaling pathways. The authors report that depleting macrophages via clodronate liposome administration alleviated kidney cell senescence and modulated the expression of GDF-15 and Klotho, suggesting these factors as potential therapeutic targets in DKD. The study presents compelling data, but certain areas would benefit from clarification and further consideration.
Major Comments
- The study examined the effects of macrophage depletion following only four weeks of clodronate liposome treatment. It is unclear whether the benefits observed are sustained over the long term or are limited to early-phase effects. Long-term depletion of macrophages could have unintended consequences, such as impaired tissue repair, heightened susceptibility to infection, or paradoxically increased inflammation.
- While the study demonstrates an association between macrophage depletion and changes in GDF-15 and Klotho expression, the mechanistic basis remains insufficiently defined. Additional experiments—such as the use of pathway inhibitors or gene silencing techniques targeting GDF-15 and Klotho—could strengthen the causal inference. Furthermore, investigating shared regulatory pathways (e.g., NF-κB or Wnt/β-catenin) could provide deeper insight into the molecular mechanisms at play.
- Albuminuria is a key marker of glomerular injury and progression in DKD. Since macrophage infiltration is reported to be linked to increased albuminuria, it would be important to assess whether clodronate liposome treatment also affected urinary albumin levels. The inclusion of such data would enhance the study's clinical relevance.
Minor Comments
- Some sentences appear disjointed or lack logical flow. For example, the transition between “GDF-15 is abundant in the placenta [14]” and “Therefore, GDF-15 may serve as a potential biomarker of DKD” is unclear and may mislead the reader. A thorough language revision is recommended to improve sentence structure and overall coherence.
- The serum creatinine levels reported in the control group (2.73 ± 0.43 mg/dL) appear considerably higher than values typically seen in rats (0.2–0.8 mg/dL). This discrepancy should be addressed, and possible reasons—such as methodological differences, assay calibration, or sample handling—should be clarified.
- Similarly, urea values reported in the study appear to be lower than expected based on established reference ranges for rats. It would be helpful for the authors to provide justification or context for these findings, possibly including reference values for their specific rat strain and measurement method.
- In the Results and Figure Legends, it should be clearly stated whether oxidative stress and inflammatory markers were measured in kidney tissue, blood, or urine. This distinction is critical for interpreting the findings correctly.
- The grouping order in the figures and tables could be improved for clarity. Presenting the groups as: (1) normal control, (2) clodronate-treated non-diabetic, (3) diabetic control, and (4) clodronate-treated diabetic, would enhance readability and facilitate comparison between treated and untreated conditions within diabetic and non-diabetic contexts.
- Measuring and reporting the effects of clodronate liposome treatment on circulating monocyte and leukocyte counts in blood would strengthen the study by confirming systemic immune modulation.
- In Figure 6, the legend describes GDF-15 and Klotho data as “stained with,” which is inaccurate for Western blot analysis. The authors should revise the description to correctly reflect that the data represent immunoblots, not histological staining.
- Ideally, Western blot results should include at least two representative bands per group to enhance reliability and allow for visual assessment of reproducibility. The current presentation in Figure 6 could be improved by including additional replicates or clarifying how the data were selected.
- The terms “DN” and “DKD” are used interchangeably throughout the manuscript. These should be defined clearly at the beginning and used consistently to avoid confusion.
Author Response
Responses to Reviewer #1’s Comments:
This study investigated the impact of macrophage depletion on kidney cell senescence in diabetic kidney disease (DKD) in a rat model, with particular emphasis on the roles of GDF-15 and Klotho signaling pathways. The authors report that depleting macrophages via clodronate liposome administration alleviated kidney cell senescence and modulated the expression of GDF-15 and Klotho, suggesting these factors as potential therapeutic targets in DKD. The study presents compelling data, but certain areas would benefit from clarification and further consideration.
Response:
We sincerely appreciate your time and effort in reviewing our manuscript and the valuable insights you provided; they have significantly contributed to the improvement of our manuscript. We have addressed your comments as follows:
Major Comments
- The study examined the effects of macrophage depletion following only four weeks of clodronate liposome treatment. It is unclear whether the benefits observed are sustained over the long term or are limited to early-phase effects. Long-term depletion of macrophages could have unintended consequences, such as impaired tissue repair, heightened susceptibility to infection, or paradoxically increased inflammation.
Response:
We appreciate your highlighting of this issue via your thoughtful comment and valuable insight. We acknowledge that our study was designed to focus on evaluating the short-term effects of macrophage depletion using clodronate liposome over a four-week period in the STZ-HFD-induced diabetic model (1). While we recognize that long-term effects, such as impaired tissue repair, increased susceptibility to infection, or heightened inflammation are important considerations (2), our study design included repeated administrations of clodronate liposome on a weekly basis (once per week for four weeks) to ensure sustained macrophage depletion during the experimental window. This approach partially mimicked a longer exposure scenario within the context of a progressive diabetic model. In addition, the STZ-HFD model used in this study mimicked the chronic metabolic stress and inflammation associated with type 2 diabetes, and our repeated dosing strategies allowed us to investigate macrophage dynamics under these pathological conditions. Nevertheless, we agree that future studies involving extended treatment duration and follow-up periods are necessary to fully elucidate the long-term safety and efficacy of macrophage duration strategies. Accordingly, we addressed this concern as both a study limitation and a future research direction in the Discussion section (page 13, lines 488–493), as follows.
“Although this study employed repeated clodronate liposome dosing within a chronic DN model, the study was limited to the assessment of the short-term effects of macrophage depletion using clodronate liposomes in DN. Accordingly, future investigations are needed to evaluate the long-term consequences and safety profile of sustained macrophage depletion, including the potential effects on tissue repair, paradoxical inflammation, and susceptibility to infection.”
References:
- Van Rooijen N, Sanders A. Liposome mediated depletion of macrophages: mechanism of action, preparation of liposomes and applications. J Immunol Methods. 1994; 174(1-2):83-93. https://doi.org/10.1016/0022-1759(94)90012-4.
- Mosser DM, Edwards JP. Exploring the full spectrum of macrophage activation. Nat Rev Immunol. 2008; 8(12):958-969. https://doi.org/10.1038/nri2448.
- While the study demonstrates an association between macrophage depletion and changes in GDF-15 and Klotho expression, the mechanistic basis remains insufficiently defined. Additional experiments—such as the use of pathway inhibitors or gene silencing techniques targeting GDF-15 and Klotho—could strengthen the causal inference. Furthermore, investigating shared regulatory pathways (e.g., NF-κB or Wnt/β-catenin) could provide deeper insight into the molecular mechanisms at play.
Response:
Thank you for your insightful and constructive comment. While our findings demonstrated a clear association between macrophage depletion and modulation of GDF-15 and Klotho expression, we acknowledge that our study did not include pathway inhibitors or gene silencing approaches to confirm the causal mechanisms involved. To the best of our knowledge, specific inhibitors or gene manipulation strategies targeting GDF-15 and Klotho are currently limited in terms of their applicability for in vivo use as we used in this model. However, we strongly agree that future studies using in vitro models or genetically modified animal models could help determine the precise molecular mechanisms and signaling pathways involved, such as NF-kB or Wnt/β catenin. A statement addressing this limitation and highlighting future research directions has been added to Discussion section (page 13–14, lines 493–498) as follows.
“Although our in vivo model revealed a clear association between macrophage depletion and the modulation of GDF-15 and Klotho expression, mechanistic experimentation using gene silencing or pathway-specific inhibitors remains limited in this context. Future studies employing in vitro models or transgenic models may help to clarify the regulatory mechanisms and confirm the causal relationships, particularly in relation to pathways such as NF-kB and Wnt/β-catenin.”
- Albuminuria is a key marker of glomerular injury and progression in DKD. Since macrophage infiltration is reported to be linked to increased albuminuria, it would be important to assess whether clodronate liposome treatment also affected urinary albumin levels. The inclusion of such data would enhance the study’s clinical relevance.
Response:
Thank you for your comment. We agree that urinary albumin would enhance the clinical relevance of the study. However, the primary aim of our study was to investigate the effect of macrophage depletion on kidney cell senescence in DN, with a specific focus on the relationship between GDF-15 and the Klotho signaling pathways. Therefore, the current study emphasized mechanistic insights at the molecular level. In addition to molecular analysis, we assessed other markers for kidney function, such as serum creatinine, urea, and BUN levels, and performed histopathological evaluation of kidney injury. Although urinary albumin was not measured in this study, we acknowledge its importance as a clinical marker. Accordingly, we added a statement to the Discussion section (page 13, lines 485–488) suggesting its inclusion in future studies, as follows.
“Future studies should consider assessing urinary albumin levels and applying functional parameters, such as the glomerular filtration rate and creatinine clearance, to enhance the clinical relevance of the findings.”
Minor Comments
- Some sentences appear disjointed or lack logical flow. For example, the transition between “GDF-15 is abundant in the placenta [14]” and “Therefore, GDF-15 may serve as a potential biomarker of DKD” is unclear and may mislead the reader. A thorough language revision is recommended to improve sentence structure and overall coherence.
Response:
We highly appreciate your review efforts and this candid observation. We revised this sentence within the manuscript to clarify it and achieve logical flow (page , line ). Notably, the entire manuscript had previously been submitted for language editing by Elsevier Language Editing Services (see the included English editing certificate). According to your recommendation, we re-checked the entire manuscript to improve sentence structure and overall coherence. If you believe that this manuscript requires further English-language editing, we are happy to oblige. However, we prefer to delay that editing process until after the reviewers have offered their complete approval of the current revisions so that it is the last step in the manuscript publication process.
- The serum creatinine levels reported in the control group (2.73 ± 0.43 mg/dL) appear considerably higher than values typically seen in rats (0.2–0.8 mg/dL). This discrepancy should be addressed, and possible reasons—such as methodological differences, assay calibration, or sample handling—should be clarified.
Response:
Thank you for this valuable comment. The current finding related to creatinine values is consistent with the published paper (PMID: 38057037, referenced below). Differences in values could be related to different animal strains, the experimental design, the calculation methods, the units of measurement, or the type of kit used. Serum creatinine levels in the current study were measured using rat colorimetric kits purchased from United Diagnostics Industry (UDi; Langenhagen, Hanover, Germany). The samples were analyzed according to the manufacturer’s instructions. The samples’ absorbance was recorded after one, two, and three minutes via the continuous use of a spectrophotometer reader (from Biochrom Ltd.; Cambridge, England, UK) at λ = 340 nm. Then, the serum creatinine levels were calculated as shown below.
CK Activity (IU/L) = = [DA/min] * 6592.
Importantly, we applied the same experimental conditions to all of the groups throughout all of the experiments, and we compared them to derive our conclusion.
Reference:
Mazani M, Mahdavifard S, Koohi A. Crocetin ameliorative effect on diabetic nephropathy in rats through a decrease in transforming growth factor-β and an increase in glyoxalase-I activity. Clin Nutr ESPEN. 2023 Dec;58:61-66. doi: 10.1016/j.clnesp.2023.08.033. Epub 2023 Sep 9. PMID: 38057037.
- Similarly, urea values reported in the study appear to be lower than expected based on established reference ranges for rats. It would be helpful for the authors to provide justification or context for these findings, possibly including reference values for their specific rat strain and measurement method.
Response:
We appreciate this comment. The current finding of urea values is consistent with the published paper (PMID: 37576066, referenced below). Differences in values could be related to different animal strains, the experimental design, the calculation methods, the units of measurement, or the type of kit used. Serum urea levels in our study were measured using rat colorimetric kits purchased from United Diagnostics Industry (UDi; Langenhagen, Hanover, Germany). The samples were analyzed according to the manufacturer’s instructions using a spectrophotometer reader (purchased from Biochrom Ltd.; Cambridge, England, UK) at λ = 630 nm. Then, the serum urea levels were calculated as follows:
Urea (mg/dl) = [Sample/Standard ] ´ 53.57
Importantly, we applied the same experimental conditions to all of the groups throughout all of the experiments, and we compared them to arrive at our conclusion.
Reference:
Alsuliam SM, Albadr NA, Alshammari GM, Almaiman SA, ElGasim Ahmed Yagoub A, Saleh A, Abdo Yahya M. Lepidium sativum alleviates diabetic nephropathy in a rat model by attenuating glucose levels, oxidative stress, and inflammation with concomitant suppression of TGF-β1. Saudi J Biol Sci. 2023 Aug;30(8):103720. doi: 10.1016/j.sjbs.2023.103720. Epub 2023 Jun 28. PMID: 37576066; PMCID: PMC10422013.
- In the Results and Figure Legends, it should be clearly stated whether oxidative stress and inflammatory markers were measured in kidney tissue, blood, or urine. This distinction is critical for interpreting the findings correctly.
Response:
Thank you for your insightful comment. This addition was made as requested; the oxidative stress and inflammatory markers were measured in kidney tissue.
- The grouping order in the figures and tables could be improved for clarity. Presenting the groups as: (1) normal control, (2) clodronate-treated non-diabetic, (3) diabetic control, and (4) clodronate-treated diabetic, would enhance readability and facilitate comparison between treated and untreated conditions within diabetic and non-diabetic contexts.
Response:
Thank you for this comment. This change was effected as requested. The grouping was reordered for the table and the following figures: Figure 1, Figure 2, Figure 3, Figure 4, and Figure 5. We maintained Figure 6 in its original form to keep it consistent with the group order in the representative blot.
- Measuring and reporting the effects of clodronate liposome treatment on circulating monocyte and leukocyte counts in blood would strengthen the study by confirming systemic immune modulation.
Response:
We highly appreciate your constructive comment and suggestion. Our study aimed to provide evidence on the impact and mechanisms of macrophage depletion on kidney tissues by utilizing clodronate liposomes as tissue-targeted therapy. We agree that presenting data on the systemic immune modulation of clodronate liposomes in the HFD/DM model might provide additional information about the systemic immunomodulatory role of clodronate liposomes in diabetes. However, this is beyond the scope of this study and could be addressed as a future research direction. Accordingly, we added the following statement to the Discussion section as a potential future research direction (page 14, lines 499–502) as follows:
“Future studies are warranted to evaluate the systemic immunomodulatory effects of clodronate liposome treatment by assessing circulating monocyte and leukocyte counts in blood, which could further validate the systemic impact of macrophage depletion beyond tissue-specific effects.”
- In Figure 6, the legend describes GDF-15 and Klotho data as “stained with,” which is inaccurate for Western blot analysis. The authors should revise the description to correctly reflect that the data represent immunoblots, not histological staining.
Response:
Thank you for this important distinction. This change was made as requested, and the sentence was rewritten as follows: “Representative immunoblot of GDF-15 antibody (a) or Klotho antibody (c).”
- Ideally, Western blot results should include at least two representative bands per group to enhance reliability and allow for visual assessment of reproducibility. The current presentation in Figure 6 could be improved by including additional replicates or clarifying how the data were selected.
Response:
Thank you for this valuable suggestion. We want to clarify that in our study, Western blotting was conducted using kidney tissue lysates from six biological replicates per group (i.e., one sample per rat, totaling six rats per group). Each immunoblot was performed independently using different kidney samples.
To enhance clarity and reduce visual complexity in the main figures, we selected one representative immunoblot per protein target per group to present in the main results (Figure 6). However, to ensure transparency and support reproducibility, we have provided the full, unprocessed immunoblot replicates as Supplementary Figure S1, which was submitted as a separate PDF file during the submission process. In addition, we would like to note that, in animal tissue lysates, it is common practice to present one representative band per group in the main figure, particularly when all biological replicates are quantified and the full blots are provided in supplementary materials. In contrast, inclusion of two bands per group is more typical in in vitro studies using cell cultures, where technical replicates are more easily controlled. Accordingly, we have added a clarification to the figure legend (page 9, lines 280–282) and the Methods section (page 17, lines 628–631).
- The terms “DN” and “DKD” are used interchangeably throughout the manuscript. These should be defined clearly at the beginning and used consistently to avoid confusion.
Response:
Thank you for this observation. We completely agree. A definition statement at the beginning of the Introduction clearly addresses this issue (page 1, lines 34–35), and we used the term consistently throughout the manuscript.
Definition statement: “Diabetic nephropathy (DN), or diabetic kidney disease (DKD), is one of the most severe complications of diabetes. It is defined as chronic kidney disease and a major cause of end-stage renal disorder in diabetic patients [1].”
Reference:
Hoogeveen, EK, The epidemiology of diabetic kidney disease. Kidney Dial. 2022 2(3):433-442.

Reviewer 2 Report
Comments and Suggestions for Authors
The manuscript presents an interesting and important subject of GDF-15 and Klotho signaling pathways on age-related diabetic kidney disease (DKD). Indeed, macrophage depletion combats kidney senescence by modulating Klotho and GDF-15 as novel targets in DN treatment.
The paper is well written throughout the text and the references are appropriate, relevant and up-to-date. However, the expression of adhesion molecules and cytokines in podocytes may be associated and interinfluenced with several growth factors (VEGF-A, TGFβ-1, FGF-23...), proteases inhibitors (TIMP-1, TIMP-2...), oxidative stress molecules (ox-LDL, Isoprostane-8 and -15), and the rapid induction of stress factors and the lack of genetic heterogeneity may not realy and completely present the intricacy and complexity of human DKD. The authors are encouraged to add in the ''Discussion'' 2-3 lines-sentences based on the current literature and their own experience concerning the association of this topic with oxidative stress-proinflammatory molecules in order to better describe this major clinical model.
Author Response
Responses to Reviewer #2’s Comments:
The manuscript presents an interesting and important subject of GDF-15 and Klotho signaling pathways on age-related diabetic kidney disease (DKD). Indeed, macrophage depletion combats kidney senescence by modulating Klotho and GDF-15 as novel targets in DN treatment.
The paper is well written throughout the text and the references are appropriate, relevant and up-to-date. However, the expression of adhesion molecules and cytokines in podocytes may be associated and interinfluenced with several growth factors (VEGF-A, TGFβ-1, FGF-23...), proteases inhibitors (TIMP-1, TIMP-2...), oxidative stress molecules (ox-LDL, Isoprostane-8 and -15), and the rapid induction of stress factors and the lack of genetic heterogeneity may not realy and completely present the intricacy and complexity of human DKD. The authors are encouraged to add in the “Discussion”' 2-3 lines-sentences based on the current literature and their own experience concerning the association of this topic with oxidative stress-proinflammatory molecules in order to better describe this major clinical model.
Response:
We sincerely appreciate your time and effort in reviewing our manuscript and the valuable insights you provided; they have significantly contributed to the improvement of our manuscript. With regard to your suggestion, we have addressed it in the Discussion section (page 13, lines 460–466) as follows:
“At the clinical level, oxidative stress-proinflammatory’s role in DN’s pathogenesis is evident (1). Additionally, an elevated level of GDF-15 predicts the mortality rate in DN patients with cardiac complications (2). In support of this fact, our findings open new avenues to explore the roles of GDF-15 and Klotho in the progression and management of high-risk DN patients. These results could be clinically translated into a multimarker informative approach in DN patients. Oxidative stress-proinflammatory combinational markers could predict the disease severity.”
References:
- Guo, W, et al., Systemic immune-inflammation index is associated with diabetic kidney disease in Type 2 diabetes mellitus patients: Evidence from NHANES 2011-2018. Frontiers in Endocrinology, 2022. Volume 13 - 2022.
- Carlsson, AC, et al., Growth differentiation factor 15 (GDF-15) is a potential biomarker of both diabetic kidney disease and future cardiovascular events in cohorts of individuals with type 2 diabetes: a proteomics approach. Ups J Med Sci. 2020. 125(1):37-43.
